# Identification of the Principle of Taste Sensors to Detect Non-Charged Bitter Substances by ^1^H-NMR Measurement

**DOI:** 10.3390/s22072592

**Published:** 2022-03-28

**Authors:** Misaki Ishida, Haruna Ide, Keishiro Arima, Zeyu Zhao, Toshiro Matsui, Kiyoshi Toko

**Affiliations:** 1Graduate School of Information Science and Electrical Engineering, Kyushu University, 744 Motooka, Nishi-ku, Fukuoka 819-0395, Japan; zeyu.zhao.720@s.kyushu-u.ac.jp; 2Department of Bioscience and Biotechnology, Faculty of Agriculture, Graduate School of Kyushu University, 744 Motooka, Nishi-ku, Fukuoka 819-0395, Japan; ide.haruna.881@s.kyushu-u.ac.jp (H.I.); k.arima@agr.kyushu-u.ac.jp (K.A.); tmatsui@agr.kyushu-u.ac.jp (T.M.); 3Research and Development Center for Five-Sense Devices, Kyushu University, 744 Motooka, Nishi-ku, Fukuoka 819-0395, Japan; toko@ed.kyushu-u.ac.jp; 4Institute for Advanced Study, Kyushu University, 744 Motooka, Nishi-ku, Fukuoka 819-0395, Japan

**Keywords:** taste sensor, NMR, caffeine, hydroxybenzoic acid, allostery, hydrogen bond, e-tongues

## Abstract

A taste sensor with lipid/polymer membranes is attracting attention as a method to evaluate taste objectively. However, due to the characteristic of detecting taste by changes in membrane potential, taste sensors cannot measure non-charged bitter substances. Many foods and medicines contain non-charged bitter substances, and it is necessary to quantify these tastes with sensors. Therefore, we have been developing taste sensors to detect bitter tastes caused by non-charged substances such as caffeine. In previous studies, a sensor for detecting bitterness caused by caffeine and theobromine, theophylline, was developed, using a membrane modified with hydroxybenzoic acid (HBA) as the sensing part. The sensor was designed to form intramolecular hydrogen bonds (H-bonds) between the hydroxy group and carboxy group of HBA and to successively cause the intermolecular H-bonds between HBA and caffeine molecules to be measured. However, whether this sensing principle is correct or not cannot be confirmed from the results of taste sensor measurements. Therefore, in this study, we explored the interaction between HBA and caffeine by ^1^H-nuclear magnetic resonance spectroscopy (NMR). By the ^1^H NMR detection, we confirmed that both the substances interact with each other. Furthermore, the nuclear Overhauser effect (NOE) of intermolecular spatial conformation in solution was measured, by which 2,6-dihydroxybenzoic acid (2,6-DHBA) preferably interacted with caffeine via the H-bonding and stacking configuration between aromatic rings. Identifying the binding form of 2,6-DHBA to caffeine was estimated to predict how the two substances interact.

## 1. Introduction

One of the methods to evaluate the taste felt by humans is to use a taste sensor or electronic tongues [1,2,3,4,5,6,7,8]. An example of the commercialized machines is TS-5000Z (Intelligent Sensor Technology, Inc., Kanagawa, Japan), which can objectively evaluate the intensity of tastes such as bitterness and sweetness by attaching a lipid/polymer membrane to the sensor electrode. The strength of each taste can be evaluated using the lipid/polymer membrane specific for each type of taste quality. The taste sensor equips multiple membranes to measure five kinds of taste qualities and is used in the development of foods and pharmaceuticals.

The change in membrane potential of the taste sensor is evaluated as the strength of the taste when taste substances interact electrically with the lipid/polymer membrane or are adsorbed onto it by hydrophobic interaction. A large number of bitter substances are included in a drink such as coffee, and the taste sensor can quantify the bitterness of coffee [9,10]. However, owing to this potentiometric measurement, conventional bitterness sensors cannot evaluate the bitterness of non-charged bitter substances such as caffeine, theobromine and theophylline included in beverages and pharmaceutical products because they cannot change the electric potential of the membrane.

To solve this problem, we developed a sensor that successfully measured the bitterness of caffeine or theobromine by modifying the surface of the membrane with hydroxybenzoic acid (HBA) [11,12,13]. The principle of the measurement considered was based on allostery, in which the intramolecular hydrogen bonds (H-bonds) are first formed between the carboxy and hydroxy groups of the HBA modified on the membrane surface. In this situation, H^+^ is dissociated from the carboxy group of HBA, which is negatively charged as a result. The intramolecular H-bonds are broken by the formation of H-bonds between the HBA and caffeine, causing H^+^ in the caffeine solution to return to the HBA and positively increasing the membrane potential [11,12]. In this principle to assume allostery, the binding of caffeine, which has no charge, results in a change in membrane potential; it suggests that taste sensors can be widely used to measure not only bitterness but also the taste caused by substances that have no charge. However, this principle is only a hypothesis at present, and we could not be sure what kind of interaction caused the change in electric potential during the detection of caffeine because taste sensors are the devices that measure only electrical changes. One of the methods to compare with the sensor outputs is ^1^H-NMR measurements.

In this study, we used ^1^H-NMR measurements to investigate the molecular interactions between HBA and caffeine. NMR measurement is a method that can analyze the molecular structure of substances in solution by observing the nuclear magnetic resonance phenomenon when atomic nuclei are placed in a magnetic field, and then has the characteristics of being able to measure solutions and analyze them non-destructively. There are different types of 1D NMR, such as ^1^H NMR and ^13^C NMR, depending on the active nucleus to be observed. The resonance frequency of NMR differs for each nuclide, but even for the same nuclide, the resonance frequency changes depending on the state of existence of each nuclide. This change is called a chemical shift, and since the electronic style of bonding contributes greatly to this shift, it is possible to analyze the state [14]. There are several types of information that can be obtained from NMR measurements, but in this study, we used chemical shifts. When a mixture of two interacting substances is measured, each spectrum shifts from the position of the spectrum when a single substance is measured because the electronic situation around the nucleus changes due to the interaction [15,16].

In this study, we also used nuclear Overhauser effect spectroscopy (NOESY). The NOESY is a two-dimensional method to obtain the spatial conformation by observing NOE between two proton nuclei in solution; the spatial distance information between the two nuclei can be obtained from the crossing peaks [17,18,19]. We used the ^1^H-^1^H NOESY method to obtain the distance information between ^1^H nuclei, which can be possible to identify atom pairs that are spatially close to each other.

By investigating whether caffeine and HBA interact in solution using ^1^H NMR measurements, we compared the results with those measured by taste sensors to confirm the correctness of the measurement principle proposed before [11,12] for measuring the bitterness caused by non-charged substances. Furthermore, by measuring NOESY of a mixed solution of caffeine and 2,6-DHBA, we confirmed which sites of the two substances were nearby during the interaction, and from this result, we predicted how the two substances interacted.

## 2. Materials and Methods

### 2.1. Reagents

We purchased dioctyl phenyl-phosphonate (DOPP), polyvinyl chloride (PVC), 2-hydroxybenzoic acid (2-HBA) and 3-trimethylsilyl-1-propanesulfonic acid-$d_{6}$ (DSS-d_6_) from FUJIFILM Wako Pure Chemical Corporation (Osaka, Japan). Tetradodecylammonium bromide (TDAB), 3,5-dihydroxybenzoic acid (3,5-DHBA), aniline and resorcinol were purchased from Sigma-Aldrich (St. Louis, MO, USA). Caffeine was obtained from Tokyo Chemical Industry Co., Ltd. (Tokyo, Japan). 2,6-Dihydroxybenzoic acid (2,6-DHBA), tartaric acid and potassium chloride (KCl) were acquired from Kanto Chemical Co., Inc. (Tokyo, Japan). Deuterium oxide (D_2_O, 99.8 atom% D) was acquired from Acros Organics (Fair Lawn, NJ, USA). Appendix A shows the chemical structures of caffeine, HBAs, aniline, and resorcinol used in this study.

### 2.2. Measuring Caffeine in Taste Sensors Using Various HBAs

In this study, we compare the response of taste sensors to caffeine using each HBA and the results of NMR measurements of mixed solutions of each HBA and caffeine. First, we measured caffeine in the taste sensor using lipid/polymer membranes modified with each HBA. The lipid/polymer membrane was made by mixing 10 mL of 3 mM TDAB in THF as a lipid, 1.5 mL DOPP as a plasticizer and 800 mg PVC as a polymer-supporting reagent. Then, the mixture solution was poured into the Petri laboratory dish (90 mm φ), and the membrane was formed because of volatilizing THF. After cutting the lipid/polymer membrane and sticking it on a sensor electrode, it was immersed in 0.3 wt% of HBA (2,6-DHBA, 2-HBA, 3,5-DHBA) solution for 48 h. Four sensor electrodes equipped with the above HBA-modified membrane were prepared and used for the measurements. The concentration of caffeine solution is 100 mM, and the solvent is a reference solution consisting of 30 mM KCl and 0.3 mM tartaric acid. When measuring, the measurement procedure was repeated four times, and the mean values and standard deviations (SDs) were calculated from n = 4 (electrode) × 4 (rotation) = 16 electrical response values. These experimental conditions were based on those of previous studies [11]. Appendix A shows a flowchart of taste sensor measurement.

### 2.3. ^1^H NMR Measurement of Caffeine and Each Substance

All ^1^H NMR spectra were obtained with an ECS-400 spectrometer (JEOL, Tokyo, Japan). In order to confirm whether caffeine interacts with each of the prepared substances, ^1^H NMR measurements were performed. The substances measured were 2,6-DHBA, 2-HBA, 3,5-DHBA, aniline, and resorcinol. Aniline is not an HBA, but it is a substance that would not interact with caffeine, so we conducted experiments to compare it with each substance. Resorcinol is similar in structure to 2,6-DHBA, which is one of the best materials for modifying the lipid/polymer membrane to obtain a large response in the taste sensor [11,12]. Therefore, the presence (in 2,6-DHBA) or absence (in resorcinol) of the carboxy group allows a comparison of the results of sensor response.

Mixed solutions of the above substances and caffeine were prepared at molar ratios of 0:1 to 3:1 (caffeine: the above substance). We prepared 9 concentrations of the mixture of caffeine and 2,6-DHBA and 4 concentrations of the mixture of caffeine and other substances for measurements. This is because the caffeine response of the taste sensor using 2,6-DHBA is the largest, and we wanted to see at what concentration ratio it starts interacting with caffeine. The solvent for each solution is D_2_O. In addition, KCl, an electrolyte, is used as a solvent for caffeine solution when measuring caffeine with taste sensors. To approach the conditions of measurement with taste sensors, 1 mM KCl was mixed in each D_2_O solution for NMR measurement.

A single pulse sequence was used for acquiring the NMR spectra employing an acquisition time of 2.73 s, a relaxation delay of 12 s and spinning at 15 Hz. The ^1^H NMR spectra were referenced to the signal of DSS-$d_{6}$ at 0.00 ppm. The DSS-$d_{6}$ solution was loaded into a 5 mm insert capillary glass tube (N-502B, Nihonseimitsu Scientific Co., Tokyo, Japan), and the tube was inserted into a 5 mm NMR sample tube (Nihinseimitsu Scientific Co.). The same DSS-loaded tube was used for all NMR measurements.

To investigate which sites of the two substances are in close proximity, we observed the NOESY spectrum of a mixed solution of 2,6-DHBA and caffeine. We prepared a mixed solution of the two substances at a molar ratio of 1:1. The concentration of each substance was 15 mM, and the solvent was D2O. The basic measurement conditions are the same as for ^1^H NMR. NOESY measurements were performed at the relaxation time of 1.2247 s, 4 scans, X points of 1024, Y points of 256, a 90° pulse-width of 12.9 µs, a relaxation delay of 2 s at 30 °C and non-spinning.

## 3. Results and Discussion

### 3.1. Response of Taste Sensor

Figure 1 shows the results of caffeine measurements using lipid/polymer membranes modified with three types of HBAs, resorcinol and aniline. The membranes modified with 2,6-DHBA showed a large response, while the moderate response appeared in the membrane modified with 2-HBA and almost no response appeared with 3,5-DHBA, resorcinol and aniline. The result of aniline is negative, but a few mV is within the error range, so this is considered no response. This is similar to the conclusion of a previous study [11], which showed that lipid/polymer membranes modified with substances that form intramolecular H-bonds are highly responsive to caffeine. This suggests that 2,6-DHBA interacts with caffeine in some way to change the membrane potential.

### 3.2. Investigation of the Interaction between HBA and Caffeine by ^1^H NMR

Figure 2 shows the ^1^H NMR spectra for each substance and caffeine in D_2_O containing 1 mM KCl. The chemical shift changes according to the mixing ratio in each substance. To make the amount of chemical shift change easier to understand, the graph is plotted with the molar ratio on the horizontal axis and the chemical shift change on the vertical axis (Figure 3). The chemical shifts of the three HBAs (2,6-DHBA, 2-HBA and 3,5-DHBA) and resorcinol change with increasing caffeine concentration. By contrast, the chemical shift of aniline did not change much with the increasing concentration of caffeine.

We compare the results of these HBAs and resorcinol with those of aniline. First of all, when the concentration is changed as in this experiment, the chemical shift may change due to the viscosity of the solution, but since there was no shift change in the aniline and caffeine solution, we can conclude that the shift changes in HBAs and resorcinol are not due to the viscosity. In addition, although aniline, HBAs and resorcinol have benzene rings, only the shift of HBAs and resorcinol changed, indicating that the shift change in these substances is not due to the π–π interaction between benzene aromatic rings of these substances and caffeine, but to other intermolecular interaction(s). Since the chemical shift of resorcinol changed, we can assume that the hydroxy group of HBA may be involved in the interaction.

### 3.3. Comparison of Taste Sensor Results and ^1^H NMR Results

Comparison of the results of the taste sensor measurements and those of ^1^H NMR suggests that the response principle proposed previously [11] is reasonable. Table 1 summarizes the results of the taste sensor shown in Figure 1 and ^1^H NMR results shown in Figure 3 for the five substances. All three HBAs and resorcinol interact with caffeine, but only the membrane modified with 2,6-DHBA shows a significant response. For the taste sensor to respond to chemical substances, the electric potential of the membrane must change; for the membrane potential to change, the taste substance, i.e., caffeine, must first interact with the membrane.

As already reported [11], HBAs with two intramolecular H-bonds (2,6-DHBA, 2,4,6-THBA) showed the response to 100 mM caffeine by over 50 mV, while HBAs with one H-bond (2-HBA, 2,5-DHBA) showed the response by 15–30 mV; HBAs with no H-bond (3,5-DHBA, 3,4-DHBA) scarcely had the response 0–10 mV. In this experiment, a large membrane potential change occurred only with 2,6-DHBA, a medium membrane potential change with 2-HBA and no membrane potential change occurred with 3,5-DHBA in accordance with the previous results [11]. 2-HBA and 3,5-DHBA surely interact with caffeine, but the membrane potential scarcely changes. This result suggests that the intramolecular H-bonds that are formed when the hydroxy group and the carboxy group of HBA are adjacent to each other are responsible for the change in membrane potential. 2,6-DHBA has two intramolecular H-bonds, 2-HBA has one, and 3,5-DHBA does not form intramolecular H-bonds. Thus, 2-HBA and 3,5-DHBA must have interacted with caffeine when measured by the taste sensor, but the magnitude of the change in membrane potential depended on the number of intramolecular H-bonds.

Whereas the change in chemical shift occurred in resorcinol, there hardly appeared a response of the resorcinol-modified sensor to caffeine. It is because resorcinol has no carboxy group to result in no formation of intramolecular H-bonds in the same way as 3,5-DHBA. Finally, aniline does not interact with caffeine, so of course, no taste sensor response was produced. Combining the taste sensor results with the ^1^H NMR results, it was shown that the intramolecular H-bonds of HBA are necessary for the taste sensor to measure the bitterness of caffeine and that the response principle proposed previously [11] was found to be more correct.

### 3.4. Prediction of the Binding form of 2,6-DHBA to Caffeine

Figure 4 shows the NOESY spectrum of a mixed solution of 2,6-DHBA and caffeine. This spectrum shows that the proton of the benzene aromatic ring in 2,6-DHBA and the proton of the methyl group in caffeine are close to each other during the interaction. Since the protons in close proximity do not interact with each other, we can assume that these protons are nearby as a result of some kinds of interaction; they are mainly H-bonds between the hydroxy group of HBA and the carbonyl group or N(imidazole) of caffeine, as discussed below.

From this result, we can predict what kind of bond is formed during the interaction. If we take into account the fact that “the hydroxy group of HBA is related to the interaction” obtained from the ^1^H NMR results and “protons are close to each other” obtained from the NOESY results, we can predict that the two substances are stacking as shown in Figure 5. This prediction agrees with the reports on cocrystals [20,21] that the hydroxy group of HBA forms an H-bond with the carbonyl group (=O) or N(imidazole) of caffeine. These results suggest that the interaction between caffeine and 2,6-DHBA is due to H-bonding between the hydroxy group of HBA and the carbonyl group or N(imidazole) of caffeine, along with π–π interaction between aromatic rings. Furthermore, since both hydroxy groups of 2,6-DHBA are involved in the binding, it is considered that 2,6-DHBA has a stronger interaction with caffeine than other HBAs.

## 4. Conclusions

In this study, we used ^1^H NMR measurements to confirm whether the principle that HBA-modified lipid/polymer membranes can detect the bitter taste of caffeine is acceptable, as suggested previously [11,12]. Measurements with taste sensors again confirmed that the response to caffeine increases with the number of intramolecular H-bonds in the HBA that modifies lipid/polymer membranes. No chemical shift of aniline when it was used as counterpart substrate strongly demonstrated that intermolecular H-bonds and π–π interaction between HBA and caffeine molecules are associated with their stable complex formation in solution. Furthermore, by comparing the results of the taste sensor with the results of ^1^H NMR, it was found that the conditions for the change in membrane potential during the measurement depended on substances that modify lipid/polymer membranes, which interact with caffeine and form intermolecular H-bonds. Based on the proximity of the sites examined by NOESY measurements, we predicted that caffeine and 2,6-DHBA are stacked together, suggesting H-bonding between the hydroxy group of HBA and the carbonyl group or -N of caffeine. This is consistent with the principle of response to caffeine in taste sensors shown previously [11,12], and we confirmed that this response principle was valid. By using NMR measurements, we were able to prove the principle of allostery—the ability to detect non-charged substances in taste sensors.

## Figures and Tables

**Figure 1 sensors-22-02592-f001:**
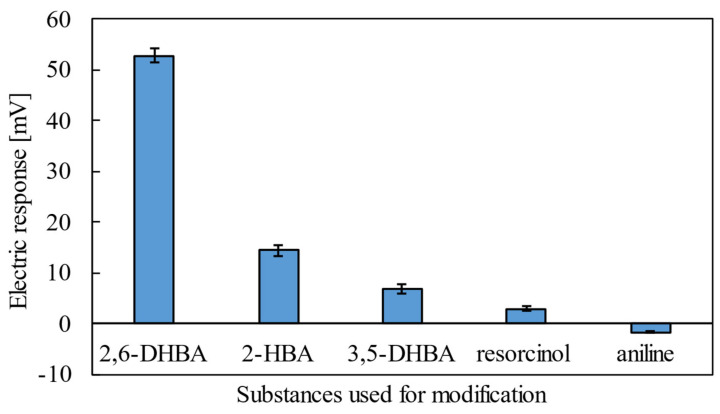
Response of taste sensor to 100 mM caffeine using the lipid/polymer membranes modified with three types of HBAs, resorcinol and aniline (the error bar expresses the SD of the data of n = 4 (electrode) × 4 (rotation) = 16 values). Lipid/polymer membranes modified with substances that form intramolecular H-bonds are highly responsive to caffeine.

**Figure 2 sensors-22-02592-f002:**
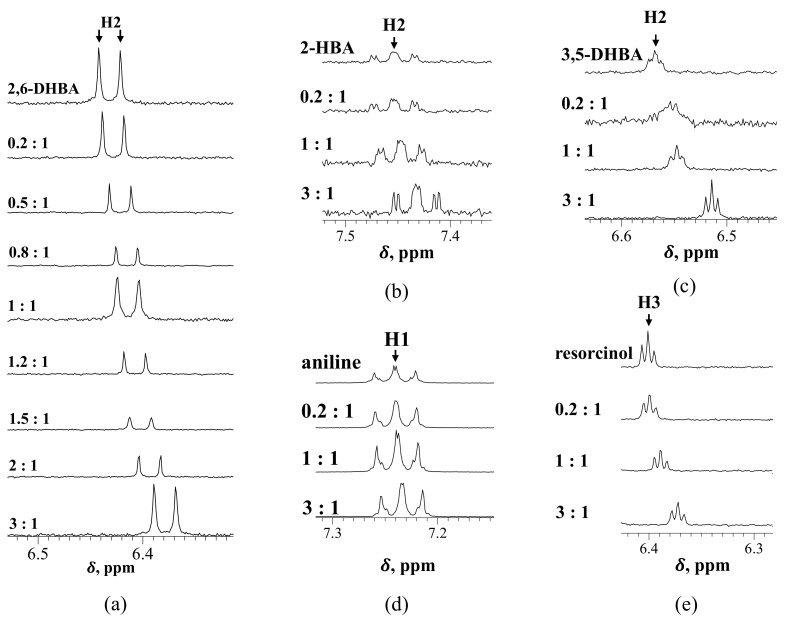
^1^H NMR spectra for each substance and caffeine: (**a**) 2,6-DHBA; (**b**) 2-HBA; (**c**) 3,5-DHBA; (**d**) aniline; (**e**) resorcinol. The chemical shifts (δ) changed with increases in molar ratio in substances except for aniline.

**Figure 3 sensors-22-02592-f003:**
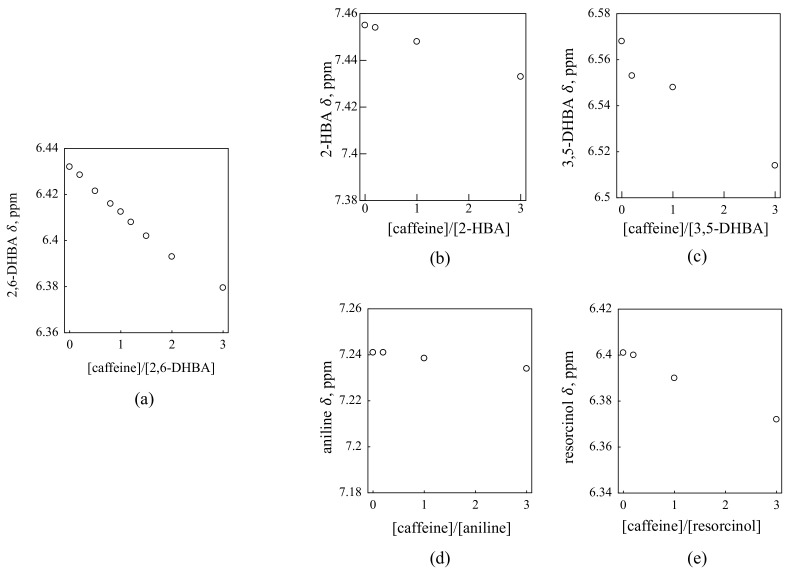
A plot of chemical shift (δ) with increasing molar ratio for each substance and caffeine: (**a**) 2,6-DHBA; (**b**) 2-HBA; (**c**) 3,5-DHBA; (**d**) aniline; (**e**) resorcinol. The chemical shifts of the three HBAs and resorcinol changed. By contrast, the chemical shift of aniline did not change. Changes in chemical shifts indicate that there is an interaction between the two substances.

**Figure 4 sensors-22-02592-f004:**
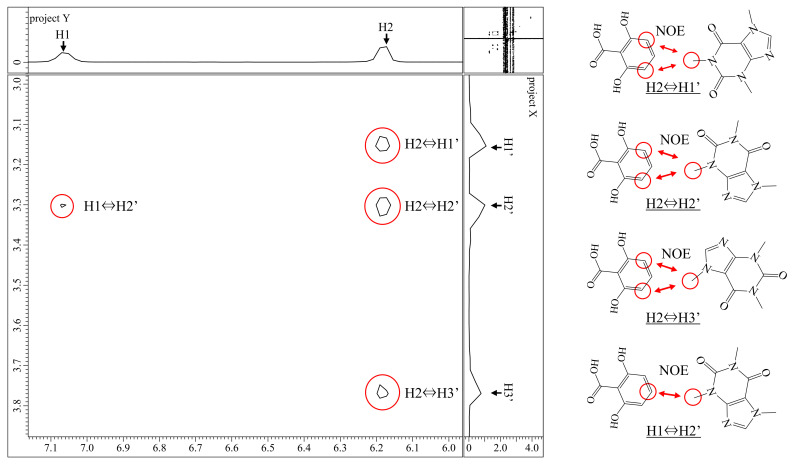
NOESY spectrum of 2,6-DHBA and caffeine and proton in proximity. Protons shown in a red circle in the right figure are close to each other as a result of some kind of interaction.

**Figure 5 sensors-22-02592-f005:**
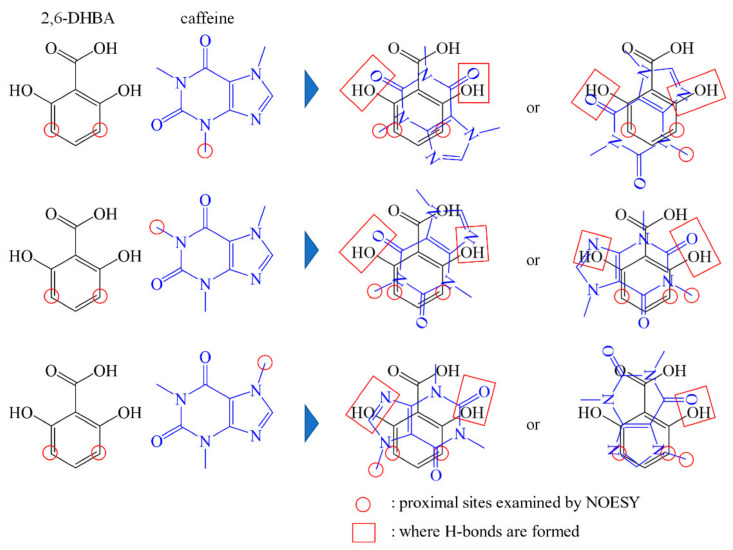
Prediction of the binding form of 2,6-DHBA and caffeine. The binding form was predicted from the information on the proton in the proximity obtained by NOESY. The red circles are the protons in close proximity obtained from NOESY. The red squares show the positions where H-bonds are formed.

**Table 1 sensors-22-02592-t001:** Summary of taste sensor results and ^1^H NMR results.

	Taste Sensor Response	Interactions Investigated by ^1^H NMR
2,6-DHBA	52 mV	Yes
2-HBA	15 mV	Yes
3,5-DHBA	7 mV	Yes
Resorcinol	3 mV	Yes
Aniline	−2 mV	No

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
