# Peer review of "Identification of the Principle of Taste Sensors to Detect Non-Charged Bitter Substances by ^1^H-NMR Measurement"

_sensors, 2022, doi:10.3390/s22072592_

Round 1
Reviewer 1 Report
A good work! By using NMR measurements, the authors clarified the detection mechanism of caffeine with HBA-modified membranes. This result maybel increase the research interests on detecting non-charged bitter substances.
Author Response
Dear Editor and Reviewer,
Thank you for the comments concerning our manuscript entitled “Identification of the Principle of Taste Sensors to Detect Non-Charged Bitter Substances by 1H-NMR Measurement”. Those comments are all valuable and very helpful for revising and improving our paper, as well as the important guiding significance to our research. We have studied comments carefully and have made corrections which we hope to meet with approval. We are submitting the corrected manuscript with the suggestion incorporated in the manuscript. The main corrections in the manuscript and the responses to the reviewer’s comments are as follows:
Reviewer #1:
A good work! By using NMR measurements, the authors clarified the detection mechanism of caffeine with HBA-modified membranes. This result maybel increase the research interests on detecting non-charged bitter substances.
Response Thank you so much for your comments. By combining the results of the taste sensor measurements with those of the NMR measurements, we were able to derive new results. We hope that there will be a lot of interest in this research.
Once again, thank you very much for your comments.
Reviewer 2 Report
The current study, “Identification of the Principle of Taste Sensors to Detect Non- 2 Charged Bitter Substances by 1H-NMR Measurement” focuses on the development of taste sensor. The study seems publishable however, the following suggested modifications would enhance the impact and usefulness of the study:
- The article needs to be revised for the grammar improvement.
- The author explains in the abstract about the limitation of taste sensors by saying that, “taste sensors cannot measure non-charged bitter substances” followed by the purpose of the current study. However, this does not comprehensively exhibit the importance of the developed sensor.
- The author in the abstract part refers to his/her previous studies, however, there is not particular reference to these studies provided.
- The abstract just talks about the Caffeine molecules to be measured. Does it mean that the sensor’s ability is limited to measuring the single substance only?
- The first figure shows the structures of reagents used in this study. The figure does not inform something unique and new. The figure either should be removed or moved to supplementary section.
- The captions of the figures are not elaborative. The captions should be modified and explained clearly and briefly to convey the information contained in the figures to users.
- The results in figure 3 does not clearly convey the information. These should be very briefly explained in the discussion section or in the figure caption.
- In the section 3.4. author states that, “This spectrum shows that the proton of the benzene aromatic ring in 2,6-DHBA and the proton of the methyl group in caffeine are close to each other during the interaction. Since the protons in proximity do not interact with each other, we can assume that these protons are nearby as a result of some kinds of interaction.” This statement should be clarified further,
- Figure 3, before conclusion looks little bit too busy. The figure should be revised and explained clearly.
- The author in the different points of study mentions the terms like NOESY etc. and the explanation for these terms is not provided. These terms should be properly explained once in the study.
Author Response
Dear Editor and Reviewer,
Thank you for the comments concerning our manuscript entitled “Identification of the Principle of Taste Sensors to Detect Non-Charged Bitter Substances by 1H-NMR Measurement”. Those comments are all valuable and very helpful for revising and improving our paper, as well as the important guiding significance to our research. We have studied comments carefully and have made corrections which we hope to meet with approval. We are submitting the corrected manuscript with the suggestion incorporated in the manuscript. The main corrections in the manuscript and the responses to the reviewer’s comments are as follows:
Reviewer #2:
- The article needs to be revised for the grammar improvement.
Response Thank you so much for your comments. We improved obvious grammar problems in the revised manuscript e.g., capital letter at the beginning of the sentence at the line 160 and figure number at the line 205.
- The author explains in the abstract about the limitation of taste sensors by saying that, “taste sensors cannot measure non-charged bitter substances” followed by the purpose of the current study. However, this does not comprehensively exhibit the importance of the developed sensor.
Response Thank you so much for your minute observation. As you mentioned, the meaning of development was not indicated. We have added the sentence explaining the significance of the development to line 19-20 in the revised manuscript.
- The author in the abstract part refers to his/her previous studies, however, there is not particular reference to these studies provided.
Response Thank you so much for your comments. We refer to previous studies in our introduction, e.g., line 56 in the revised manuscript.
- The abstract just talks about the Caffeine molecules to be measured. Does it mean that the sensor’s ability is limited to measuring the single substance only?
Response Thank you so much for valuable comments. As you mentioned, this study focuses on caffeine. This is because caffeine is one of the most common non-charged bitter substances. This sensor, developed in a previous study, has been shown to measure theophylline and theobromine in addition to caffeine. We have added some text and references to line 48-53 in the revised manuscript to make them easier to understand.
- The first figure shows the structures of reagents used in this study. The figure does not inform something unique and new. The figure either should be removed or moved to supplementary section.
Response Thank you very much for your valuable suggestion. This figure has been moved to the supplemental materials in the revised manuscript (line 284-288)
- The captions of the figures are not elaborative. The captions should be modified and explained clearly and briefly to convey the information contained in the figures to users.
Response Thank you very much for your valuable suggestion. We have added figure explanations to make it easier for the user to understand in the revised manuscript. (line 171-174, 186-187, 189-192, 248-249, 263-265)
- The results in figure 3 does not clearly convey the information. These should be very briefly
explained in the discussion section or in the figure caption.
Response Thank you very much for your valuable suggestion. We added a brief explanation in the caption in the revised manuscript (line 189-191).
- In the section 3.4. author states that, “This spectrum shows that the proton of the benzene aromatic ring in 2,6-DHBA and the proton of the methyl group in caffeine are close to each other during the interaction. Since the protons in proximity do not interact with each other, we can assume that these protons are nearby as a result of some kinds of interaction.” This statement should be clarified further,
Response Thank you very much for your comments. We added text to line 241-244 to clearly explain the results in the revised manuscript.
- Figure 3, before conclusion looks little bit too busy. The figure should be revised and explained clearly.
Response Thank you very much for your valuable suggestion. We divided the figure into two parts to make the results easier to understand in the revised manuscript. (Figure2 and Figure 3)
- The author in the different points of study mentions the terms like NOESY etc. and the explanation for these terms is not provided. These terms should be properly explained once in the study.
Response Thank you very much for your comments. Our definition of NOESY is described in line 84-89.
We tried our best to improve the manuscript and made some changes in the manuscript. We appreciate for Editors/Reviewers’ warm work earnestly and hope that the correction will meet with approval.
Once again, thank you very much for your comments and suggestions.
Reviewer 3 Report
Dear Authors,
In general, the manuscript is good to read, the structure of the work is clear and has a sufficient literature review. I present my comments below:
- Lines 35-36: Smell is responsible for 80-90 percent of the taste. The taste buds play a much smaller role in this process - 10 - 20 percent. The final taste impression is mainly determined by the volatile substances released by the chemicals contained in the food. We inhale odor molecules through the front nostrils, but also through the rear nostrils, which transport volatile substances generated during chewing to the nose.
- Lines 59-61: What other reference tests can be used to compare with the electrical signal results from the sensors?
- Lines 107-108: What does the measuring stand look like? Maybe a flowchart should be attached to supplementary materials?
- Line 157. The capital letter at the beginning of the sentence should be.
- Figure numbers are repeated and are not quoted everywhere in the text (figure lines 166).
- Figure 2 line 166. Why does aniline have negative signal values?
- Table 1 is partly a repetition of the information in Figure 1 (line 166). The sentence under Table 1 should be a line below.
- Lines 246-248: What about measuring caffeine in coffee? The content is 1 to 3%. Will the sensors generate different current signals, depending on the content?
Author Response
Dear Editor and Reviewer,
Thank you for your comments concerning our manuscript entitled “Identification of the Principle of Taste Sensors to Detect Non-Charged Bitter Substances by 1H-NMR Measurement”. Those comments are all valuable and very helpful for revising and improving our paper, as well as the important guiding significance to our research. We have studied comments carefully and have made a correction which we hope to meet with approval. We are submitting the corrected manuscript with the suggestion incorporated in the manuscript. The main corrections in the manuscript and the responses to the reviewer’s comments are as follows:
Reviewer #3:
- Lines 35-36: Smell is responsible for 80-90 percent of the taste. The taste buds play a much smaller role in this process - 10 - 20 percent. The final taste impression is mainly determined by the volatile substances released by the chemicals contained in the food. We inhale odor molecules through the front nostrils, but also through the rear nostrils, which transport volatile substances generated during chewing to the nose.
Response Thank you so much for your comments. As you point out, it is true that taste felt in the brain is largely affected by smell owing to volatile substances. The contribution of smell is very large. Among this situation, we consider that the role of e-tongues or taste sensors is to objectively evaluate the taste reception occurring at the gustatory cell without taking into account the effect of smell. So, researchers concerning with e-tongues or taste sensors including us should integrate the output of e-tongues with that of e-noses. Whereas such approaches have been reported so far, we consider promoting them more. We appreciate your kind, precise comments.
- Lines 59-61: What other reference tests can be used to compare with the electrical signal results from the sensors?
Response Thank you so much for your comments. We are considering 1H NMR as a test that can be used to compare. We have added a sentence indicating this to line 68-69 in the revised manuscript.
- Lines 107-108: What does the measuring stand look like? Maybe a flowchart should be attached to supplementary materials?
Response Thank you very much for your valuable suggestion. We have added a measurement flowchart to the supplemental material in the revised manuscript. (line 289-297)
- Line 157. The capital letter at the beginning of the sentence should be.
Response Thank you so much for your minute observation. A few words were missing and have been corrected in the revised manuscript. (line 160, 205)
- Figure numbers are repeated and are not quoted everywhere in the text (figure lines 166)
Response Thank you so much for your minute observation. A word in line 160 was missing and has been corrected in the revised manuscript.
- Figure 2 line 166. Why does aniline have negative signal values?
Response Thank you very much for your comments. The output of a few mV is an error range, which sometime appears due to the small difference in characteristics of four sensor electrodes, the small changes in the composition of reference and washing solutions during the repetitive measurements, and so on. We have added the sentence to explain this to line 164-165 in the revised manuscript.
- Table 1 is partly a repetition of the information in Figure 1 (line 166). The sentence under Table 1 should be a line below.
Response Thank you very much for your valuable suggestion. As you mentioned, the data is repeated. We have added text to line 205-207 for clarity and have left space between the table and the text to make it easier to read in the revised manuscript. (line 214-215)
- Lines 246-248: What about measuring caffeine in coffee? The content is 1 to 3%. Will the sensors generate different current signals, depending on the content?
Response Thank you very much for your comments. Coffee contains a variety of bitter substances and a small amount of caffeine. For other substances, it has been found that bitterness due to charged bitter substances can be measured with a bitter taste sensor for charged substances. The sensor used in this study has also been found to be selective for non-charged bitter substances. There is a paper that measured the bitterness of coffee (Wu, X.; et al. Quantification of Bitterness of Coffee in the Presence of High-Potency Sweeteners Using Taste Sensors. Sensors Actuators, B Chem. 2020, 309, doi:10.1016/j.snb.2020.127784.). Coffee contains many bitter substances that are below the threshold for human taste perception, and they act together to give a bitter taste. Therefore, the contribution of caffeine to taste is low, but the non-charged bitter substances as a whole do contribute. This research began with the fact that taste sensors can respond to charged bitter substances but not to non-charged bitter substances. We have added some text and references to line 48-53 to make this fact easier to understand.
We tried our best to improve the manuscript and made some changes in the manuscript. We appreciate for Editors/Reviewers’ warm work earnestly and hope that the correction will meet with approval.
Once again, thank you very much for your comments and suggestions.
Round 2
Reviewer 2 Report
The authors have now incorporated the comments, therefore I recommend its acceptance.